# A Theory of Equivalence-Preserving Program Embeddings

## Abstract

Program embeddings are used to solve tasks such as *code clone detection* and *semantic labeling*. Solutions to these *semantic tasks* should be invariant to semantics-preserving program transformations. When a program embedding function satisfies this invariance, we call it an *equivalence-preserving program embedding function*. We say a programming language can be *tractably embedded* when we can construct an equivalence-preserving program embedding function that executes in polynomial time in program/input length and produces program embeddings that are proportional to the input length. Determining whether a programming language can be tractably embedded is the *equivalence-preserving program embedding problem*. We formalize this problem and theoretically characterize when programming languages can be tractably embedded. To validate our theoretical results, we use the BERT-Tiny model to learn an equivalence-preserving program embedding function for a programming language that can be tractably embedded and show the model fails to construct an equivalence-preserving program embedding function for a similar language that is intractable to embed.

## 1 Introduction

Emerging research demonstrates that powerful new techniques can solve challenging program reasoning tasks, such as *code clone detection* and *semantic labeling* (Hu et al., 2017; Mou et al., 2016). At the core of many techniques lie *program embeddings*, fixed-size representations produced from program text that model a program property. For example, a program embedding may be a vector of floating-point numbers that models a program's input-output behavior. A common and effective method for producing program embeddings is to use a neural network (Yu et al., 2019; Ben-Nun et al., 2018). While the empirical results of these techniques have been demonstrated, there has been, to date, little theoretical understanding of the capabilities of program embedding techniques.

**Semantic Tasks.** A key first step towards understanding program embeddings is understanding the tasks they are designed to solve. In this work, we focus on *semantic tasks*, tasks that depend only on the input-output behavior of a program. Equivalently, the solution to a semantic task is invariant to all semantics-preserving transformations on the input program.

**A Theory of Equivalence-Preserving Program Embeddings.** When a program embedding technique is invariant to semantics-preserving transformations, two programs' embeddings are identical exactly when the programs are semantically equivalent. We call such a technique an *equivalence-preserving program embedding function*. We say a programming language can be *tractably embedded* when we can construct an equivalence-preserving program embedding function that runs in time polynomial in program/input length and the embedding size is proportional to input length. The problem of determining whether a programming language can be tractably embedded is the *equivalence-preserving program embedding problem*. To date, this problem has not been identified in the literature, and it has been unknown under what conditions it can be solved. We provide the first theoretical characterization of this problem by proving necessary and sufficient conditions for when a programming language can be tractably embedded.

**Empirical Study.** We apply our theory to programming languages for modular addition and bitvector arithmetic (Bosselaers et al., 1994; Barrett et al., 1998). We prove the modular addition language can be tractably embedded, while the bitvector arithmetic language cannot. We find that a BERT-Tiny model can learn an equivalence-preserving embedding function for the modular addition language, but not the bitwise arithmetic language (Bhargava et al., 2021; Turc et al., 2019). We also validate that as the number of possible inputs increases for these languages, the model maintains 100% train and test accuracy for the modular addition language, but accuracy degrades quickly for the bitvector arithmetic language.

Our contributions are as follows:

1. We define the equivalence-preserving program embedding problem and identify applications of equivalence-preserving program embeddings.

2. We prove necessary and sufficient conditions for when a programming language can be tractably embedded.

3. We hypothesize that programming languages that can be tractably embedded are easier to learn than those that are intractable to embed and provide evidence with an empirical study of languages for modular addition and bitvector arithmetic.

We present definitions and results constituting the first steps in a theory of equivalence-preserving program embeddings. Building on these foundations, future work can better analyze existing programming languages, design new programming languages that can be tractably embedded, and develop principled approximations for programming languages that cannot be tractably embedded.

## 2 Equivalence-Preserving Program Embeddings In Practice

Semantic tasks are those where only a program's input-output behavior must be modeled. We show how equivalence-preserving program embeddings can solve these tasks and briefly explain how researchers solve them.

**Code Clone Detection.** Code clone detection is the task of identifying whether a pair of programs are equivalent and is often used to find duplicate code in a codebase or identify software vulnerabilities in programs (Hu et al., 2017). Given an equivalence-preserving program embedding function, one can solve this problem by embedding both programs and checking equivalence. Researchers produce program embeddings that solve this problem via neural networks (Yu et al., 2019), execution of programs at a fixed set of inputs (Hu et al., 2017), and locality-sensitive hashing of such executions (Pewny et al., 2015).

**Semantic Labeling.** In semantic labeling, the goal is to label a program as having a particular semantic property among a fixed-size collection of possible properties. Examples include determining the algorithm a program implements and identifying whether a program exhibits a particular error. An equivalence-preserving program embedding function maps each program to its label, which identifies the program's functionality. Note that the labels come from a fixed set representing all program functionalities of interest. Researchers solve this problem using neural program embeddings (Mou et al., 2016; Ben-Nun et al., 2018; Puri et al., 2021; Wang et al., 2018).

## 3 A Theory of Equivalence-Preserving Program Embeddings

In this section, we formalize the equivalence-preserving program embedding problem and prove conditions under which it can be solved.

We define a programming language $\mathcal{P}$ as a set of strings with additional structure. For one, deciding membership of programs in a programming language is often decidable (e.g., by parsing and type checking), meaning one can enumerate all programs from shortest to

longest[12]. Furthermore, every programming language can be ascribed a denotation function $[\![\cdot]\!]$ that maps each program $p$ to its meaning $[\![p]\!] : \mathcal{I} \to \mathcal{O}$ as a mathematical function from inputs to outputs (Scott, 1977). We provide a full definition in Appendix A.

We model embeddings as elements of a finite set, from which a fixed-size bit-vector representation can be recovered. A *program embedding function* is a function $e : \mathcal{P} \to \mathcal{E}$ that maps a program in $\mathcal{P}$ to an element of a finite set $\mathcal{E}$. A *program embedding* is an element $e(p)$ produced by applying a program embedding function $e$ to a program $p \in \mathcal{P}$.

To define equivalence-preserving program embeddings, we need a notion of program equivalence. Because programs denote mathematical functions, we can define program equivalence in terms of *functional equivalence*. Programs $p, q \in \mathcal{P}$ are functionally equivalent ($[\![p]\!] = [\![q]\!]$) when for every input $x \in \mathcal{I}$, $[\![p]\!](x) = [\![q]\!](x)$, where $=$ is an equivalence relation defined over $\mathcal{O}$.

**Definition 3.1.** A program embedding function $e : \mathcal{P} \to \mathcal{E}$ is *equivalence-preserving* if for every $p, q \in \mathcal{P}$,

$$e(p) = e(q) \Leftrightarrow [\![p]\!] = [\![q]\!]. \tag{1}$$

In other words, program embeddings should be equivalent precisely when the programs denote equivalent functions.

We say a programming language can be tractably embedded when there exists a polynomial-time, equivalence-preserving embedding function (in terms of program length $|p|$ and input bits $\log|\mathcal{I}|$) and the embedding size scales with $\log|\mathcal{I}|$. The latter condition prevents encoding the entire input-output behavior of the program (see Corollary A.6).

**Definition 3.2.** A programming language $\mathcal{P}$ can be *tractably embedded* when:

1. One can construct an equivalence-preserving embedding function $e$ such that, for all $p \in \mathcal{P}$, the time complexity of $e$ is polynomial in the length of the input program $|p|$ and the minimum description length of the input space $\log|\mathcal{I}|$ (i.e., it runs in $O(\text{poly}(|p|, \log|\mathcal{I}|))$ time).

2. The length of the embedding's bit-level representation $|e(p)|$ is $O(\log|\mathcal{I}|)$ for all $p \in \mathcal{P}$.

We are now ready to define the equivalence-preserving program embedding problem.

**Definition 3.3.** The *equivalence-preserving program embedding problem* is, given a programming language, determine whether it can be tractably embedded.

While this problem sits at the core of many techniques to solve semantic tasks, it cannot always be solved. We can tractably embed a programming language when there are polynomially many *semantic equivalence classes* and an efficient *canonicalization* procedure. A semantic equivalence class for a program $p \in \mathcal{P}$ is the set $\{q \mid [\![q]\!] = [\![p]\!]\}$. We define a canonicalizer $c$ as a function from programs to programs such that programs $p, q \in \mathcal{P}$ canonicalize to the same program exactly when they are semantically equivalent. That is,

$$[\![p]\!] = [\![q]\!] \Leftrightarrow c(p) = c(q).$$

**Theorem 3.1.** A programming language $\mathcal{P}$ can be tractably embedded exactly when there are $O(\text{poly}(|\mathcal{I}|))$ semantic equivalence classes and there is a canonicalizer that runs in $O(\text{poly}(|p|, \log(|\mathcal{I}|)))$ time for every program $p \in \mathcal{P}$.

*Proof.* $\Rightarrow$) By condition (2) of Definition 3.2, there are $O(\log|\mathcal{I}|)$ bits in the embedding and therefore $O(\text{poly}(|\mathcal{I}|))$ distinct semantic equivalence classes. An equivalence-preserving embedding function gives decidable semantic equality, so we use Lemma A.3 to collect the shortest representative program for every semantic equivalence class. We then build a mapping from each program embedding to its corresponding canonical form, taking $O(\text{poly}|\mathcal{I}|)$ space and having $O(1)$ time lookup via perfect hashing. To canonicalize a program $p$, we embed

---

[1] To enumerate all programs, enumerate all strings in the set of symbols for the language, then use the membership decision procedure to determine which strings to output.

[2] Note that assuming decidable membership means our results exclude some complex languages such as dependently typed ones, which only admit semidecidable membership.

it (in $O(\text{poly}(|p|, \log |\mathcal{I}|))$ time), then map it to the appropriate canonical form in constant time, and output the canonical form in $O(|p|)$ time.

$\Leftarrow$) We build an equivalence-preserving program embedding function by running the canonicalizer and then mapping from canonical forms to embeddings. By Lemma A.3, we collect the shortest representative programs from each semantic equivalence class into a list. We then create a mapping from each canonical form to a bitstring encoding of its index, giving $O(1)$ lookup time, similarly to above. The program embedding function satisfies property (1) of Definition 3.2 because the canonicalizer runs in $O(\text{poly}(|p|, \log |\mathcal{I}|))$ time and mapping a canonical form to its embedding takes constant time. It satisfies property (2) since there are $O(\text{poly}(|\mathcal{I}|))$ equivalence classes, meaning there are $O(\log |\mathcal{I}|)$ bits in the embedding. $\square$

**Implications.** While the equivalence-preserving program embedding problem is undecidable in general, our results precisely describe the properties of programming languages for which the problem is not only decidable, but also tractable. In Appendix A, we provide an extended theory that describes the properties of programming languages for which the embedding merely exists or is computable, which are both weaker guarantees than tractability.

## 4 EMPIRICAL STUDY

Our theory demonstrates the conditions under which a programming language is tractably embeddable. However, not all programming languages are tractably embeddable and yet many approaches apply a variety of techniques, including neural networks, to produce embeddings for programs from languages for which the equivalence-preserving embedding problem is intractable. Thus, there is a fundamental question: what is the observable difference between applying a program embedding technique to a tractably embeddable programming language versus one that is not?

In this section, we construct and study the behavior of a neural program embedding function applied to programs from two programming languages: one that can be tractably embedded, while the other cannot. Our results show that the test accuracy of our neural embedding function on the equivalence-preserving program embedding problem degrades quickly — as the input space size grows — for the intractable language whereas performance remains at 100% for the tractable language.

### 4.1 A TRACTABLE PROGRAMMING LANGUAGE OF MODULAR ADDITION

We first consider a programming language with modular addition in a single variable. This is a subset of modular arithmetic languages that arise in algorithms for computer algebra, cryptography, and error-correcting codes (Bosselaers et al., 1994; Giorgi et al., 2009; Hoeven et al., 2016). The grammar is:

$$e := \texttt{0} \mid \texttt{1} \mid \texttt{x} \mid e \texttt{ + } e$$

The denotation function for terms in this language is parameterized by the modulus $n \in \mathbb{Z}^+$:

$$\llbracket \texttt{0} \rrbracket_n = 0$$
$$\llbracket \texttt{1} \rrbracket_n = 1$$
$$\llbracket \texttt{x} \rrbracket_n = x$$
$$\llbracket e \texttt{ + } e \rrbracket_n = (\llbracket e \rrbracket_n + \llbracket e \rrbracket_n) \mod n$$

where $x$ is an integer variable satisfying $0 \leq x < n$.

**Theorem 4.1.** The modular addition language can be tractably embedded.

*Proof.* In Appendix B.1. $\square$

### 4.2 AN INTRACTABLE PROGRAMMING LANGUAGE OF BITVECTOR ARITHMETIC

We extend the tractable language from Section 4.1 to support a subset of bitvector arithmetic, resulting in a programming language that is intractable to embed. Bitvector arithmetic

languages are used for the verification and synthesis of low-level code (Barrett et al., 1998; Jha et al., 2010; Inala et al., 2016). The grammar for the subset we consider is:

$$e := 0 \mid 1 \mid \texttt{x} \mid e \texttt{ + } e \mid e \texttt{ \& } e \mid e \texttt{ | } e \mid \texttt{\~}e$$

with additional denotations for logical conjunction $\wedge$, disjunction $\vee$, and negation $\neg$:

$$\cdots$$
$$[\![e \texttt{ \& } e]\!]_n = ([\![e]\!]_n \wedge [\![e]\!]_n) \mod n$$
$$[\![e \texttt{ | } e]\!]_n = ([\![e]\!]_n \vee [\![e]\!]_n) \mod n$$
$$[\![\texttt{\~}e]\!]_n = (\neg [\![e]\!]_n) \mod n$$

where $x$ is an integer variable satisfying $0 \leq x < n$.

**Theorem 4.2.** If $P \neq NP$, the bitvector arithmetic language cannot be tractably embedded.

*Proof.* In Appendix B.2. $\square$

### 4.3 METHODOLOGY

We use the BERT-Tiny architecture to learn a program embedding function by phrasing the embedding problem as a supervised classification problem: given the syntax of a program $p \in \mathcal{P}$, predict its semantic equivalence class (Bhargava et al., 2021; Turc et al., 2019). When the network achieves 100% train and test accuracy, we say it has learned an equivalence-preserving program embedding function. See Appendix C.1 for details.

Because BERT-Tiny is a feedforward model, it can perform a fixed amount of computation. As we vary the input space size, both languages will reach a point where the required amount of computation for an equivalence-preserving program embedding function will exceed that available to the architecture. Thus, beyond some point, the network must approximate and we hypothesize that accuracy for the intractable language will diminish more quickly than for the tractable language.

#### 4.3.1 VARYING INPUT SPACE SIZE

To vary the input space size, we first note that the tractable and intractable programming languages have denotations parametric in the modulus $n$. Increasing $n$ increases the number of values the variable $x$ ranges over, meaning it increases the input space size $|\mathcal{I}|$. Consequently, increasing $n$ increases the number of semantically distinct program denotations. For example, in the tractable language $[\![\texttt{x}]\!]_2 = [\![\texttt{1 + 1 + x}]\!]_2$, but $[\![\texttt{x}]\!]_3 \neq [\![\texttt{1 + 1 + x}]\!]_3$, and as a result, there are more semantic equivalence classes modulo 3.

#### 4.3.2 UNIFYING SYNTAX

We first considered constructing the dataset by generating programs from the grammar of each programming language. However, using different grammars for the two languages introduces a methodological concern: there are fewer operators in the tractable language than in the intractable language, so low accuracy could be explained by the need to learn embeddings for more operators.

To address this concern, we use the same program syntax for both the tractable and intractable language but interpret the two differently. For the tractable language, all unary operators in the intractable language are treated as no-ops and all binary operators are treated as addition. For example, $x \vee y$ is interpreted as a bitwise OR for the intractable language and as $x + y$ in the tractable language.

#### 4.3.3 DATASET GENERATION

To generate a dataset, we sample 500,000 abstract syntax trees (ASTs) with a fixed number of nodes, then select a subset with a balanced number of examples for each semantic equivalence class. For example, the expression `((~0 + x) | x)` has 6 nodes. In our empirical study, we use ASTs with 11 nodes (see Appendix C.2 for justification).

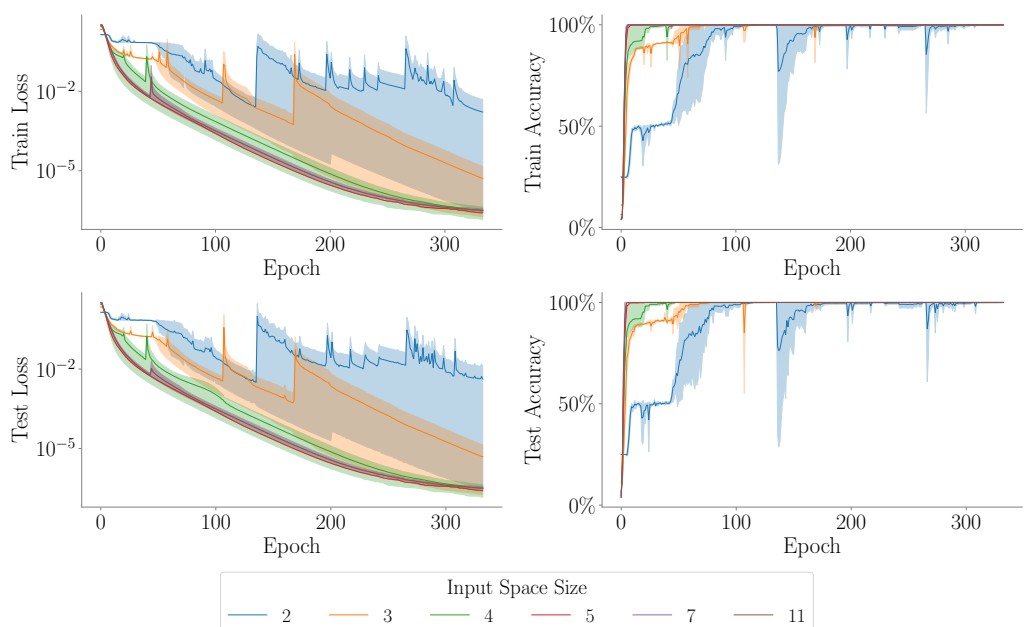

Figure 1: Average loss and accuracy graphs for the tractable language. Shaded areas show the minimum and maximum values across trials. Almost every configuration achieves 100% test accuracy in $\leq 133$ epochs, demonstrating that BERT-Tiny can learn an equivalence-preserving program embedding function for the tractable language on the input space sizes we evaluated on.

In the generated dataset, the label for a program is a label corresponding to a semantic equivalence class. We identify the semantic equivalence class by enumerating all inputs $x \in \mathbb{Z}_m$, where $m$ is the modulus, and recording the outputs. See Appendix C.3 for details.

## 4.4 EXPERIMENTS

We sweep over input space sizes in the set $\{2, 3, 4, 5, 7, 11\}$ for both the tractable and intractable languages and train BERT-Tiny to learn an equivalence-preserving program embedding function. We chose this set of input space sizes because it forms a sequence where the number of semantic equivalence classes is monotonically increasing for both languages. See Appendix C.4 for details.

## 4.5 RESULTS

Figures 1 and 2 present the losses and accuracies from training BERT-Tiny to learn an equivalence-preserving program embedding function for the tractable and intractable languages, respectively. Figure 3 distills these results into plots showing average accuracy vs. input space size for both languages.

**Tractable.** Figure 1 shows BERT-Tiny learns an equivalence-preserving embedding function for the tractable language at every input space size we evaluated on. That is, BERT-Tiny achieves 100% train and test accuracy on every configuration, and moreover, the two track strongly throughout training.

Interestingly, as $|\mathcal{I}|$ increases, the network learns an equivalence-preserving embedding function in fewer epochs. For example, when $|\mathcal{I}| = 2$, the network requires $\approx 100$ epochs to reach 100% test accuracy, but when $|\mathcal{I}| = 11$, it requires 5 epochs.

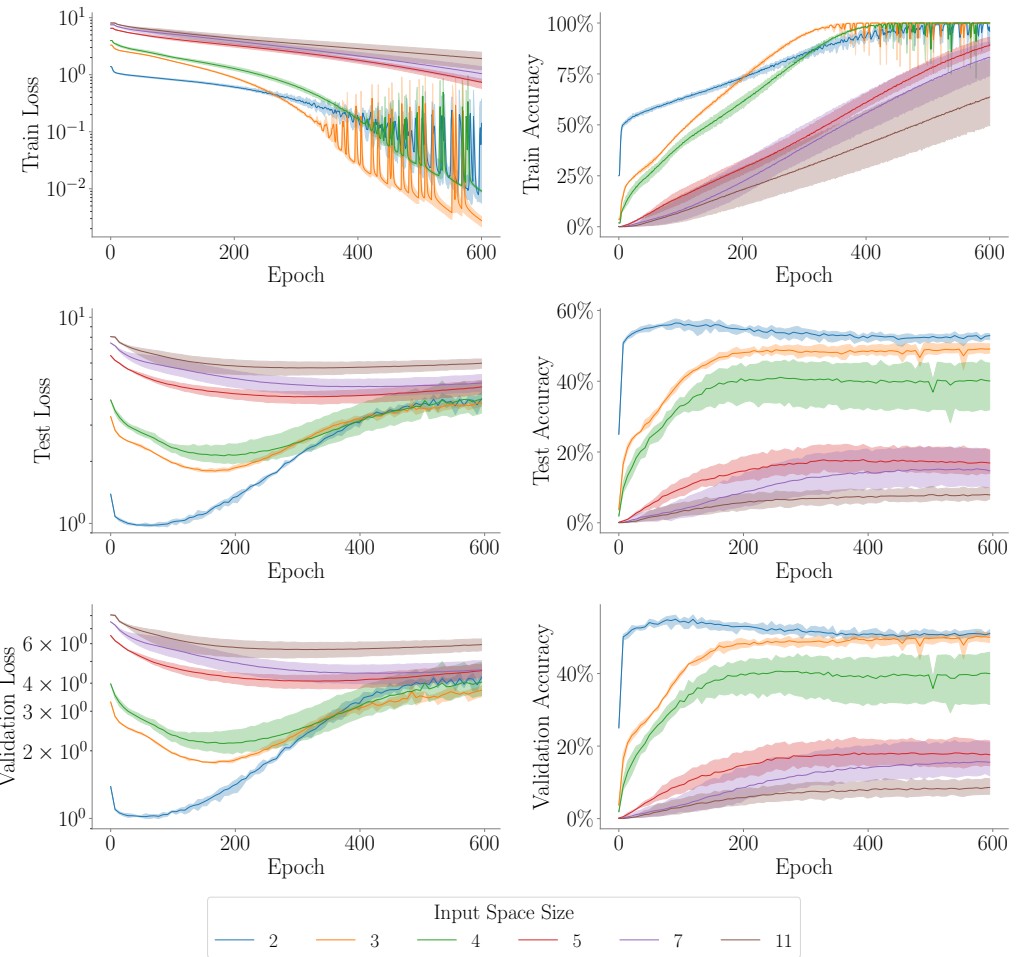

Figure 2: Average loss and accuracy graphs for the intractable language. Shaded areas show the minimum and maximum values across trials. After 600 epochs, no configuration achieves 100% train and test accuracy, demonstrating that BERT-Tiny is unable to learn an equivalence-preserving program embedding function for the intractable language on the input space sizes we evaluated on.

**Intractable.** Figure 2 shows BERT-Tiny was unable to learn an equivalence-preserving program embedding function for the intractable language on any input space size we evaluated on. While some configurations with input space size $|\mathcal{I}| \leq 4$ achieve near-100% train accuracy, no configuration achieves greater than 55% test accuracy, and average test accuracy decreases monotonically in the input space size. For $|\mathcal{I}| > 4$, average train accuracy trends downward, reaching $\approx 65\%$ at $|\mathcal{I}| = 11$.

For $|\mathcal{I}| \in \{2, 3, 4\}$, we observe that the loss and accuracy begin to rapidly oscillate between nearly 100% and as low as 73%, while validation and test remain relatively stable.

**Analysis.** Figure 3 shows the tractable language achieves 100% train and test accuracy on all input space sizes we evaluated on, while test accuracy is never 100% for the intractable language and it trends downward as the input space size increases. Thus, as expected, test accuracy deteriorates more quickly for the intractable language as we increase the input space size, compared to the tractable language.

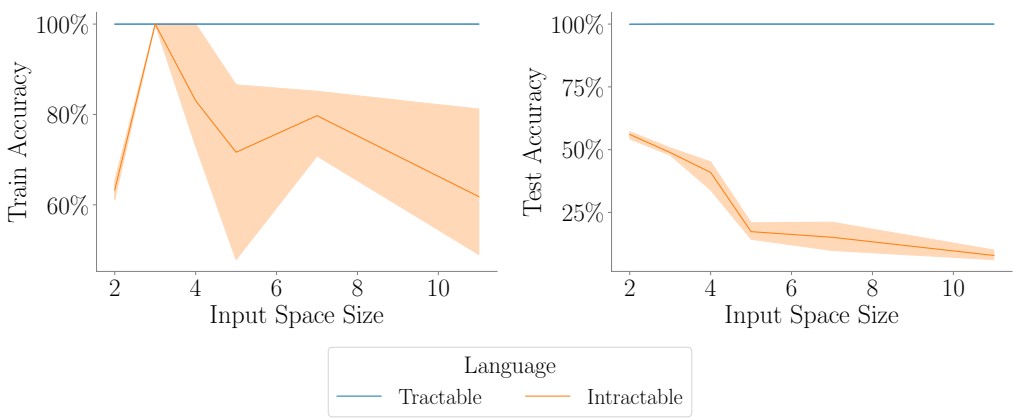

Figure 3: Average train and test accuracy vs. input space size for the tractable and intractable languages. We collect the accuracy for each trial at the epoch with the highest validation accuracy. These results show BERT-Tiny is able to learn an equivalence-preserving program embedding function for the tractable language at every input space size we evaluated on, while it was unable to learn an equivalence-preserving program embedding function for the intractable language at any input space size we evaluated on.

## 5 Discussion

In this section, we identify opportunities for future work including relaxations of the equivalence-preserving program embedding problem, program embeddings for modeling performance, and distance-preserving program embeddings.

### 5.1 Approximate Equivalence-Preserving Program Embeddings

Our results show that even commonly used subsets of basic-block assembly code (Section 4.2) cannot be tractably embedded. We discuss approaches that approximate equivalence-preserving program embeddings by either approximating programming language semantics or approximating equivalence preservation.

**Subsetting the Language.** A programming language that is intractable to embed may contain a subset that can be tractably embedded, in which case, we can restrict our consideration to this subset. For example, the modular addition programming language in Section 4.1 is a subset of the bitvector arithmetic programming language in Section 4.2; the former can be tractably embedded, while the latter cannot.

**Approximating Semantics.** When a programming language cannot be tractably embedded, we may approximate its semantics. For example, the language of Turing machine descriptions that denote 1 if the Turing machine halts on all inputs and 0 otherwise is not computable and therefore not tractable to embed. If we change the denotation function to be 1 if every input halts after a fixed number of iterations and 0 otherwise, then there is a computable, equivalence-preserving program embedding function for this programming language. Similarly, a programming language with unbounded loops is not tractably embeddable, so we may approximate its semantics by modeling only a fixed number of iterations.

**Almost Equivalence Preserving Program Embeddings.** If we have a metric on denotations of programs, we can relax equivalence preservation by enforcing that when two programs have equivalent embeddings, they are semantically similar (i.e., $e(p) = e(q) \Rightarrow d([\![p]\!], [\![q]\!]) < \epsilon$), and furthermore, $e$ is the embedding function with the smallest image

among all other feasible embedding functions.[3] For example, a programming language with rational constants in the unit interval $[0, 1]$ has no equivalence-preserving program embedding function because there are infinitely many semantic equivalence classes. However, we can build an almost equivalence-preserving with absolute error bounded by $1/2^{k-1}$. It maps a given program to an index corresponding to the nearest of $k$ uniformly spaced rationals between 0 and 1 (inclusive). To use the original denotation, we can modify the specification to encode preferences for some programs (e.g., smaller constants) by defining a distribution over programs, then improving the approximation for more important programs.[4]

**Probably Approximately Equivalence-Preserving Program Embeddings.** A *probably approximately equivalence-preserving program embedding function* maps similar programs to the same embedding with some probability. Locality-sensitive hashing can be used to satisfy this specification, because it guarantees that, for some $c > 1$, $R > 0$, and probabilities $P_1, P_2$, if $d(\llbracket p \rrbracket, \llbracket q \rrbracket) \leq R$, then $\Pr[e(p) = e(q)] \geq P_1$, and if $d(\llbracket p \rrbracket, \llbracket q \rrbracket) \geq cR$, then $\Pr[e(p) = e(q)] \leq P_2$. For example, Pewny et al. (2015) solve cross-architecture bug search (a form of code clone detection) using an embedding comprised of many input-output pairs that are compressed using locality-sensitive hashing, for faster comparisons.

## 5.2 DISTANCE-PRESERVING PROGRAM EMBEDDINGS

An even stronger condition than equivalence preservation is distance preservation (i.e., isometry). This corresponds to the desideratum that the embeddings of similar programs are close (Wang et al., 2020; Peng et al., 2021; Alon et al., 2019). Expressing this condition requires a metric on denotations of programs and program embeddings. A distance-preserving program embedding function could be useful for variants of semantic code search, where the task is not only to find semantically identical programs, but also semantically similar programs.

## 5.3 MODELING NON-STANDARD SEMANTICS

One can model intensional behavior of a program (on its own or in addition to semantic behavior), such as its size, its control flow structure, or its execution cost. For example, embeddings that capture the performance of a program could be formalized by supplying a denotation function that, instead of modeling functional equivalence, counts the number of steps the program executes, averaged over all inputs. Such an embedding would distinguish between, for example, quicksort and bubblesort. One can interpret Mendis et al. (2019) as building a program embedding function that models performance of basic block assembly code to predict throughput, a metric of performance.

## 6 CONCLUSIONS

In this paper, we develop the first steps towards a theory of program embeddings by narrowing our focus to programming embeddings that are used in semantic tasks, tasks that depend only on the input-output behavior of the program. For such tasks, an ideal embedding technique is equivalence-preserving: two programs' embeddings are identical exactly when the programs are semantically equivalent.

Our results define the equivalence-preserving program embedding problem and provide both necessary and sufficient conditions for when a programming language can be tractably embedded. Our empirical study shows that neural networks are able to more accurately embed a tractable language than an intractable language.

Taken together, our work holds out the promise of a future, complete understanding of the fundamental tension between programming language expressivity and embeddability, along with new directions for the co-design of languages and embedding techniques.

---

[3] This last condition is to prevent degenerate solutions where every program is mapped to a distinct embedding.

[4] Floating-point numbers can be seen as approximations of rationals in this sense.

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

# A    EXTENDED THEORY

We first provide a formal definition of programming language.

**Definition A.1.** A *programming language* $\mathcal{P} = \langle \mathcal{L}, \mathcal{L}_{(\cdot)}, [\![\cdot]\!] \rangle$ is a tuple of:

- A language $\mathcal{L} \subseteq \Sigma^*$ over an alphabet of symbols $\Sigma$

- An enumeration order $\mathcal{L}_{(\cdot)} : \mathbb{N} \to \mathcal{L}$ satisfying the conditions of computable enumerability (i.e., for every element $p \in \mathcal{L}$, there exists a unique $n \in \mathbb{N}$ such that $\mathcal{L}_n = p$) and such that $|\mathcal{L}_n| \leq |\mathcal{L}_{n+1}|$ for all $n \in \mathbb{N}$.

- A denotation function $[\![\cdot]\!] : \mathcal{L} \to (\mathcal{I} \to \mathcal{O})$ mapping programs to mathematical functions, where the input space is $\mathcal{I}$ and the output space $\mathcal{O}$ is also equipped with an equivalence relation $=$ decidable in $O(\log |\mathcal{O}|)$ time.

For convenience, we write $p \in \mathcal{P}$ to mean $p \in \mathcal{L}$.

## A.1    EXISTENCE

In this section, we give necessary and sufficient conditions for an equivalence-preserving embedding function to exist.

**Theorem A.1.** An equivalence-preserving embedding function exists for a programming language $\mathcal{P} = \langle \mathcal{L}, \mathcal{L}_{(-)}, [\![\cdot]\!] \rangle$ exactly when there are finitely many semantic equivalence classes.

*Proof.* $\Rightarrow$) The contrapositive is that every language with infinitely many semantic equivalence classes does not have an equivalence-preserving embedding function. Suppose there were such an equivalence-preserving embedding function into a set with $n$ elements. Then, given $n + 1$ semantically distinct programs, two distinct programs must have the same embedding by the pigeonhole principle, so the embedding function is not equivalence-preserving. Contradiction.

$\Leftarrow$) We show that finitely many semantic equivalence classes implies the existence of an equivalence-preserving embedding function. Choose $[\![\cdot]\!]$ as the embedding function. Since there are finitely many semantic equivalence classes, $[\![\mathcal{L}]\!]$ is a finite set[5], meaning it is indeed an embedding. By definition, programs are functionally equivalent exactly when their denotations are equal, so this embedding preserves equivalences as well.    $\square$

The example below is a programming language that cannot be embedded.

**Example A.2.** The programming language of integers (e.g., -100, 1, 5 are all programs) has no embedding because each number forms its own semantic equivalence class and there are infinitely many numbers. More generally, languages with infinite datatypes such as linked lists and binary trees have no embedding.

## A.2    COMPUTABILITY

The embedding given by Theorem A.1 is not always useful because it may not be computable, so we need a stronger condition to ensure computability.

The equivalence-preserving embedding function we develop in this section requires representative programs for each semantic equivalence class in a programming language. The following lemma provides a procedure to identify them.

**Lemma A.3.** When a programming language $\mathcal{P} = \langle \mathcal{L}, \mathcal{L}_{(-)}, [\![\cdot]\!] \rangle$ has $n$ semantic equivalence classes and semantic equality is decidable, there is an algorithm that collects representative programs $p_1, \ldots, p_n$ for each semantic equivalence class in finite time.

*Proof.* We enumerate through $\mathcal{L}$, maintaining a set of representatives $\mathcal{R}$ for each semantic equivalence class seen so far. For each program $p$ we encounter, we check $[\![p]\!] = [\![r]\!]$ for all

---

[5] $[\![\mathcal{L}]\!]$ is notation for the image of the programming language under the denotation function (i.e,. $\{[\![p]\!] \mid p \in \mathcal{L}\}$)

$r \in \mathcal{R}$. If $p$ is inequivalent to all such $r$, then it represents a new semantic equivalence class, and we add $p$ to $\mathcal{R}$. Once $|\mathcal{R}| = n$, we have a representative for each semantic equivalence class, so we terminate the procedure.

Now, we show the search for representatives terminates. Let $\mathcal{C} = \left\{ \llbracket e \rrbracket^{-1} \mid e \in \llbracket \mathcal{L} \rrbracket \right\}$ be a partition of $\mathcal{L}$ into semantic equivalence classes. Each semantic equivalence class $C \in \mathcal{C}$ has a member $\mathcal{L}_i \in C$ that occurs before all other $C' \in \mathcal{C}$ in the enumeration order and it does so in finite time, by computable enumerability. The algorithm will terminate after encountering a representative from each class, so it will terminate after encountering the last representative in the enumeration order. That is, it will terminate after

$$\max \left\{ \underset{i, \llbracket \mathcal{L}_i \rrbracket \in C}{\arg \min} \mathcal{L}_i \;\middle|\; C \in \mathcal{C} \right\}$$

iterations. This quantity is finite, because it is the maximum of a collection of finite values, so the algorithm terminates in finite time. $\qquad \square$

Now, with the lemma above, we show decidable semantic equality is the only additional property required to have necessary and sufficient conditions for a programming language to be computably embedded.

**Theorem A.4.** There is a computable, equivalence-preserving embedding for a programming language $\mathcal{P}$ exactly when there are finitely many semantic equivalence classes and semantic equality is decidable in $\mathcal{P}$ (for every $p, q \in \mathcal{P}$, $\llbracket p \rrbracket = \llbracket q \rrbracket$ is decidable).

*Proof.* $\Rightarrow$) Suppose we have a computable, equivalence-preserving embedding $e$ for $\mathcal{P}$. By Theorem A.1, $\mathcal{P}$ must have finitely many semantic equivalence classes. Given programs $p, q \in \mathcal{P}$, we can decide semantic equality by checking $e(p) = e(q)$, since $e$ is computable and equality of embeddings is computable.

$\Leftarrow$) Suppose $\mathcal{P}$ has $n$ semantic equivalence classes and decidable semantic equality. By Lemma A.3, we can find representative programs $p_1, \ldots, p_n$ in finite time. Then, we embed a given program $p$ by identifying the $p_i$ such that $\llbracket p \rrbracket = \llbracket p_i \rrbracket$ (using decidable equality of programs), and setting the embedding $e(p)$ to be $i$.

$\qquad \square$

The following programming language can be embedded, but cannot be computably embedded.

**Example A.5.** Consider the programming language of Turing machine descriptions that denote 1 if the Turing machine halts on all inputs and 0 otherwise. There are only two semantic equivalence classes, so a program embedding function exists. However, it is not computable because semantic equality is not decidable by a reduction from the Halting problem.

**Corollary A.6.** There is a computable equivalence-preserving embedding for programming languages with finite input and output spaces.

*Proof.* Let $\mathcal{P}$ be a programming language with finite input and output spaces $\mathcal{I}, \mathcal{O}$. There are $|\mathcal{O}|^{|\mathcal{I}|}$ possible denotations of functions in $P$. Let $\mathcal{Q}$ be a language of encodings of these functions (e.g., programs in $\mathcal{Q}$ could be sets of every input-output pair). Since the construction of $\mathcal{Q}$ is computable and it satisfies the conditions of Theorem A.4, we can construct a computable equivalence-preserving embedding. We can then embed a program in $\mathcal{P}$ by running it on every input, recording outputs, to write a program in $\mathcal{Q}$ that we embed. $\qquad \square$

By Corollary A.6, bounded datatypes over bounded data such as all fixed-length lists of bounded integers have computable equivalence-preserving embedding functions.

## B  Tractability Proofs

### B.1  A Tractable Programming Language of Modular Addition

**Theorem 4.1**  The programming language of modular addition is tractable. In particular, there are $n^2$ semantic equivalence classes.

*Proof.* Every expression in the language denotes an expression of the form $kx + c$, where $0 \leq k < n$ and $0 \leq c < n$, and each $(k, c)$ pair indexes into a distinct function signature. Thus, there are at most $n^2$ semantic equivalence classes.

To show there are at least $n^2$ equiv classes, we show that each pair is unique. All of the equations below are mod $n$. When $kx + c_1 = kx + c_2$ where $c_1 \neq c_2$, then $c_1 = c_2$, contradiction. Suppose $k_1 x + c = k_2 x + c$ where $k_1 \neq k_2$, then $(k_2 - k_1)x = 0$, but at $x = 1$, this means that $k_1 = k_2$, contradiction. When $k_1 x + c_1 = k_2 x + c_2$ where $k_1 \neq k_2$ and $c_1 \neq c_2$, then $(k_2 - k_1)x = c_1 - c_2$, but at $x = 0$, this means $c_1 = c_2$, contradiction.

Thus, there are precisely $n^2$ semantic equivalence classes. Evaluating the function at 0 and 1 gives us the coefficients $(k, c)$ and takes time linear in the program size and constant in the input space. $\qquad\square$

### B.2  An Intractable Programming Language of Modular Addition and Bitwise Logic

**Theorem 4.2**  If P $\neq$ NP, the programming language of arithmetic and bitwise logic is intractable.

*Proof.* Suppose we have a tractable embedding function $e$ for $\mathcal{P}$. We solve boolean satisfiability by encoding a circuit as a program $p$ and outputting SAT if and only if $e(p) \neq e(0)$. Therefore, we have a polynomial-time decision procedure for SAT, which is NP-complete, and P = NP. Contradiction. $\qquad\square$

## C  Extended Methodology

### C.1  Learning an Embedding Function

To learn an equivalence-preserving embedding function, we train a transformer neural network to classify program text to a fixed set so that equivalent programs are mapped to the same element of that set.

| | |
|---|---|
| Dataset size $|\mathcal{D}|$ | 18000 |
| Train set size | 12000 |
| Test set size | 3000 |
| Validation set size | 3000 |
| Number of epochs (tractable) | 333 |
| Number of epochs (intractable) | 600 |
| Learning rate $\eta$ | 1e-4 |
| Batch size $|B|$ | 128 |
| AST size | 11 |
| Number of AST Samples | 500,000 |
| Trials per configuration[6] | 3 |

**Model.**  Kanade et al. (2020) showed that the BERT (large) is capable of learning program embeddings that perform well across a number of programming language processing tasks. Since our dataset is small when compared to the dataset size that BERT is trained on, we opt for a smaller model: BERT-Tiny (Bhargava et al., 2021; Turc et al., 2019). We use the `BertForSequenceClassification` model from the `transformers` library (Wolf et al., 2020). We remove all dropout from the model. Table C.1 shows the hyperparameters for our setup.

**Training.** We train the model on a standard cross-entropy loss using the Adam optimizer with a learning rate of 1e-4. We shuffle the dataset $\mathcal{D}$ before each epoch, and we use a batch size of 128. For the tractable language, we train for 333 epochs. For the intractable language, this is not long enough to converge on all configurations, so we train for 600 epochs.

**System.** We run all Transformer experiments on an NVIDIA Tesla V100 GPU. For the tractable language, each configuration takes about 20 minutes. For the intractable language, each configuration takes about 2 hours.

## C.2 CHOICE OF AST SIZE

Since we want a fixed program length (meaning a fixed AST size), we want to show a fixed AST size does not obscure the relationship between the input space size and the number of semantic equivalence classes. Figure 7 shows that, for both languages, the relationship between input space size and number of semantic equivalence classes is preserved as we vary the AST size. In particular, this means if we included all AST sizes less than a given bound, we would have the same relationship between the input space size and the number of semantic equivalence classes. Thus, choosing a fixed AST size of 11 does not threaten our results in this regard.

## C.3 DATASET GENERATION

We first generate all ASTs of size 11 (Appendix C.3.2) and label them, then extract a balanced subset (Appendix C.3.1).

For the tractable language, we generate datasets with 15000 examples each, and we use an 80/20 train/test split. For the intractable language, not all runs achieve 100% test accuracy, so we generate a validation set as well. That is, we generate datasets with 18000 examples each, and we use a 60/20/20 train/test/validation split. Note that due to the balanced-class constraint in data generation, we generate a different dataset for each input space size and language.

### C.3.1 BALANCED DATASET GENERATION

The distribution of equivalence classes in a random sample of generated ASTs is far from uniform. The equivalence classes for functions $\lambda x. 0$, $\lambda x. 1$, and $\lambda x. x$ are represented disproportionately often ($\approx 17\%$, $\approx 7\%$, and $\approx 7\%$, respectively, at various moduli $m \geq 4$) and a long tail of infrequently occurring classes.

If we considered all ASTs, we might find enough examples for most classes to build a balanced dataset, but at larger AST sizes, the set of all ASTs is quite large. For example, at size 11, there are 22,240,092 ASTs. Enumerating this set and determining the equivalence class of each would take prohibitively long. So we need to balance the number of classes $c$ we pick for the dataset and the number of examples per class $x$ so the following equation is satisfied: $c \cdot x = |\mathcal{D}|$.

Figure 4 plots equivalence classes, ordered by frequency, on the $x$ axis and the number of expressions in an equivalence class on the $y$ axis. Navigating the balance of $c$ and $x$ amounts to finding a rectangle in this plot with area $|\mathcal{D}|$. If we wanted to, for example, have $\geq 5$ examples per class (the brown line in the plot), we can see that increasing the number of samples allows us to include more equivalence classes. Note that the strategy of finding rectangles means we may underrepresent the true number of classes at a given modulus.

For each dataset we generate, we use the shortest rectangle with at least 6 examples per class. That is, we want as many classes as possible, but we want at least 6 examples per class (4 for training, 1 for test, and 1 for validation).

To generate the final dataset, we sample 500,000 ASTs and find a rectangle satisfying the constraints above. Figure 5 shows that using 500,000 AST samples across all input space sizes provides datasets with distributions of semantic equivalence classes that match the

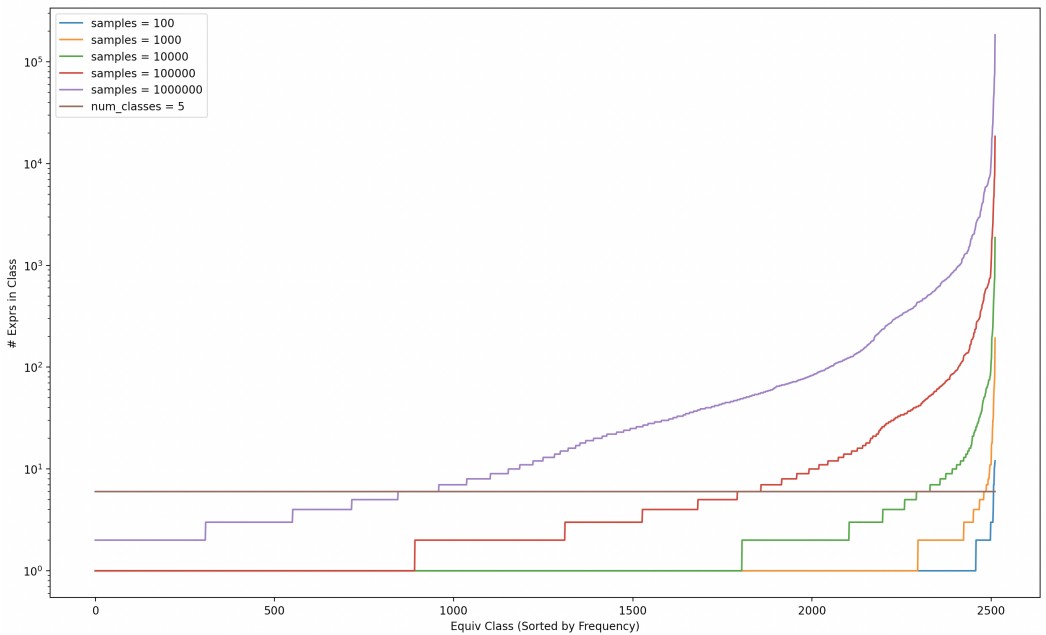

Figure 4: Semantic equivalence classes sorted by the number of expressions in the equivalence class.

Table 1: Number of semantic equivalence classes in dataset generated at each input space size

| Input Space Size | 2 | 3 | 4 | 5 | 7 | 11 |
|---|---|---|---|---|---|---|
| # Tractable Classes | 4 | 9 | 16 | 22 | 26 | 26 |
| # Intractable Classes | 4 | 27 | 53 | 834 | 2143 | 3000 |

distributions of classes encountered during sampling. Table 1 shows the number of semantic equivalence classes at each input space size in the final dataset.

### C.3.2 AST GENERATION

We define *AST size* as the number of nodes in an AST. We generate ASTs of a fixed size exhaustively by dynamic programming. Every generated program is valid (syntactically and semantically). Our code also caches enumerated expressions of each expression size. To generate an AST of size $n$, at the end of execution, all ASTs of size 1 to $n$ are stored in the cache and each unique expression of size $n$ is generated precisely once.

The base case is a constant or variable of integration. For an AST of size $n$, the algorithm generates all ASTs of size $n-1$ and feeds each to the unary not function. Similarly, for every binary operation the algorithm sets the left and right children to every possible AST pair coming from the partition of $n-1$ objects. Figure 6 provides the algoirthm described above in Python syntax.

### C.4 CHOICE OF INPUT SPACE SIZE

The relationship between the input space size and the number of semantic equivalence classes for the tractable language, meaning we could choose any increasing sequence of input space sizes to answer our research question for the tractable language. However, the relation between the input space size and the number of semantic equivalence classes in the intractable language is complicated. In particular, it is not monotonic, but there

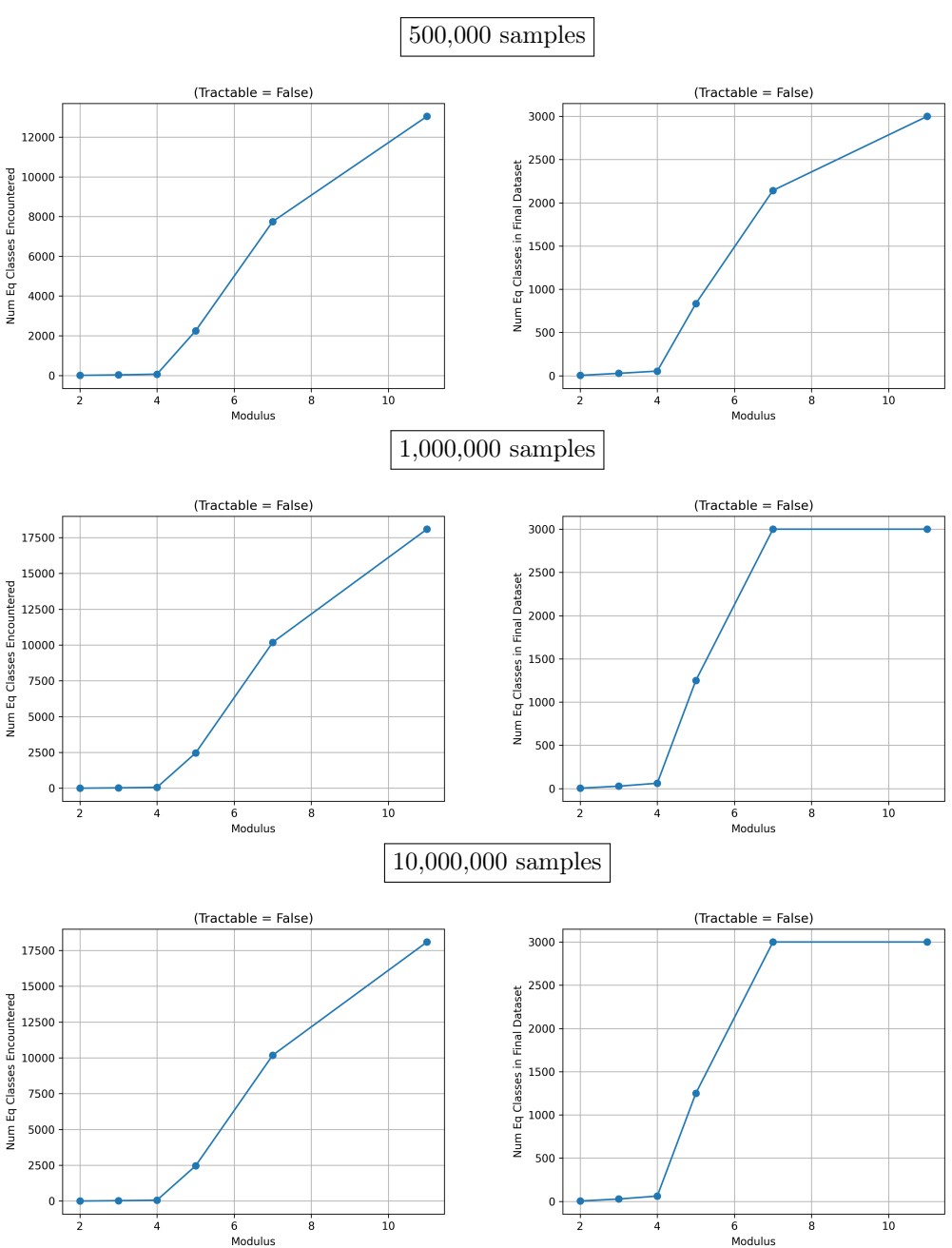

Figure 5: Comparison of the number of semantic equivalence classes encountered during sampling vs. the number of semantic equivalence classes included in the final dataset. We make this comparison for sample sizes of 500,000, 1 million, and 10 million ASTs, and we find that using 500,000 samples gives the number of encountered classes that matches most closely with the number of classes in the final dataset.

```
def enum_ast(size: int) -> list[Expr]:
    if size == 1:
        return [i for i in CONSTS] + [Var("x")]

    # Unary trees
    trees = [Not(e) for e in enum_ast(size - 1)]

    # Binary trees
    for i in range(1, size - 1):
        for binop in BINOPS:
            for left in enum_ast(i):
                for right in enum_ast(size - i - 1):
                    trees.append(binop(left, right))
    return trees
```

Figure 6: The data generation algorithm to enumerate ASTs of a fixed size without caching.

exist monotone subsequences. Figure 7 shows the relationship for both languages. For our empirical study, we identify $\{2, 3, 4, 5, 7, 11\}$ as a sequence that provides a monotonically increasing number of semantic equivalence classes for both languages.

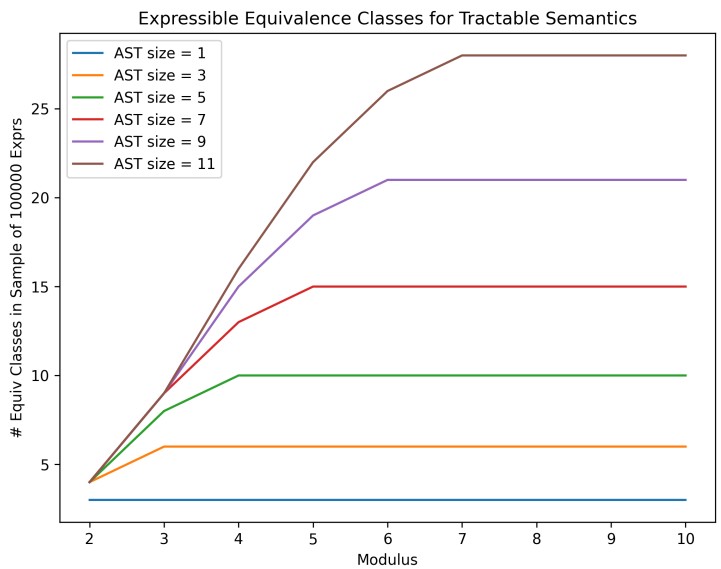

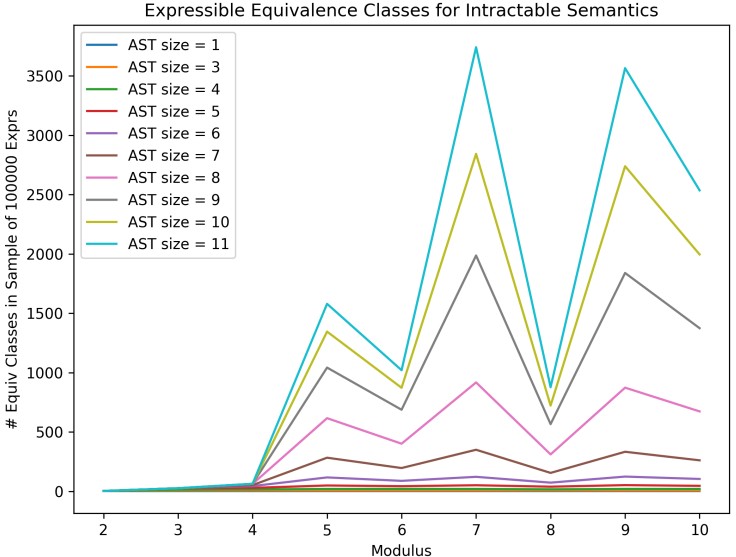

Figure 7: Number of semantic equivalence classes encountered in 100,000 AST samples vs. input space size. We plot this relationship for the tractable language in Section 4.1 (Top) and the intractable language in Section 4.2 (Bottom) and for AST sizes from 1 through 11. **(Top)** For the tractable language, we only see new semantic equivalence classes at odd AST sizes, since every operator is binary. At each AST size, as we increase the input space size, the number of semantic equivalence classes increases monotonically until plateauing. **(Bottom)** For the intractable language, we do not see a monotonic relationship between the input space size and the number of semantic equivalence classes, for any AST size. However, we do see that the relative relationship across input space sizes is preserved as we vary the AST size.

