# OpenReview forum: "A Theory of Equivalence-Preserving Program Embeddings"
_ICLR.cc/2023/Conference — Submitted to ICLR 2023_

### Official Review · Reviewer_Pnio · 2022-10-23

**Confidence:** 4
**Correctness:** 3
**Technical Novelty And Significance:** 1
**Empirical Novelty And Significance:** 2
**Recommendation:** 3

**Clarity, Quality, Novelty And Reproducibility:**

**Clarity**

In the empirical study, the authors present two toy languages. The semantics of bitvector arithmetic is not defined for logical symbols. For example, what is 2 & 3 mod 7? It is fine to omit the semantic specification for modular addition because it has natural mathematical semantics, but bitvector arithmetic needs such specification.

**Quality and Novelty**

 In essence, this paper studies the equivalence problem, which is a classical formal language problem. It appears that the authors are unaware of those results, and propose an ad-hoc theory. Therefore, the quality and novelty are insignificant at this time.

I think that understanding the hardness of semantic decidability can be an interesting angle in understanding semantic tasks arising from deep learning models. However, more works need to be done. For example, for existing programming semantic tasks, can the authors classify their hardness using the theory developed in this paper?

**Strength And Weaknesses:**

**Strength**

Analyzing the hardness of equivalence decidability can be an interesting perspective to understand the learning capacity of neural network models, especially for programming language tasks.

**Weakness**
1. The theory is ad hoc:
The (semantic) equivalence problem is a classical problem and has been studied for decades for different formal systems, for example, automata, regular expressions, and pushdown systems. They are often at least PSPACE-hard. These are naturally encountered formal systems. In this paper, the authors impose assumptions like the (denotational) semantics is from a finite set, the input space is finite. Those assumptions in general do not hold.

2. The techniques are elementary:
The techniques used to develop the theory are elementary. With the finiteness and some efficient procedure assumptions, the proofs are usually direct manipulations of definitions and assumptions. It does not seem that there are useful theoretical or mathematical techniques.

3. The evaluation is rough:
The evaluations are on two toy languages defined in this paper, and the input space size is at most 11. It is conceivable that modular addition is easier than bitvector arithmetic to learn because the latter is a direct extension of the former. It is inconclusive to understand model's capacity to learn the semantics just based on this toy evaluation.

**Summary Of The Paper:**

This paper aims to provide a theory to analyze the semantic equivalence of programming languages. This is the classical equivalence problem and in general undecidable. This paper additionally imposes some assumptions and studies when the equivalence can be efficiently decided, defined as tractably embedded in this paper. The authors also conduct an empirical study, aiming to show that BERT-Tiny can learn languages with tractably embedded semantics but not intractably embedded semantics.

**Summary Of The Review:**

This paper presents a theory to study the semantic equivalence of programming languages. However, the theory is ad hoc and the evaluation is inconclusive. This paper is not ready for acceptance yet.

---

> ### Author Response · Authors · 2022-11-18
> **Reviewer Pnio Response**
>
>
> > The theory is ad hoc: The (semantic) equivalence problem is a classical problem and has been studied for decades for different formal systems, for example, automata, regular expressions, and pushdown systems. They are often at least PSPACE-hard. These are naturally encountered formal systems.
>
> We are aware of the equivalence problem in complexity theory. We are not aware of (and the reviewer has not provided any) results that provide a formal characterization of the hardness of the equivalence-preserving embedding problem. One key difference in the equivalence-preserving embedding problem that hinders the applicability of prior results is that an embedding must be a fixed-size representation. For example, regular expressions can be efficiently canonicalized but have an infinite number of semantic equivalence classes, precluding the possibility of an equivalence-preserving embedding function.
>
> We are also not aware of any work that characterizes the number of semantic equivalence classes or the time complexity of the equivalence problem in terms of the input space size (i.e., the number of unique strings, say, regular expressions are subject to). If such work exists, we would be excited to read it as well.
>
> > The techniques are elementary: The techniques used to develop the theory are elementary. With the finiteness and some efficient procedure assumptions, the proofs are usually direct manipulations of definitions and assumptions. It does not seem that there are useful theoretical or mathematical techniques.
>
> The novelty of the paper is in the conceptual framework, definition of tractability for program embedding, and theorems. Developing new proof techniques is not a contribution of the paper.
>
> > The evaluation is rough: The evaluations are on two toy languages defined in this paper, and the input space size is at most 11. It is conceivable that modular addition is easier than bitvector arithmetic to learn because the latter is a direct extension of the former. It is inconclusive to understand model's capacity to learn the semantics just based on this toy evaluation.
>
> In our methodology, the neural network trains on the same program syntax for both the tractable and intractable languages (Section 4.3.2). The interpretation of the languages is what varies. That is, the only difference in the training data is the labels. The fact that this particular semantic extension leads to a dramatic decrease in performance reflects the importance of studying tractability, because we control for confounders such as syntactic differences or imbalanced equivalence classes.
>
> > The semantics of bitvector arithmetic is not defined for logical symbols. For example, what is 2 & 3 mod 7? It is fine to omit the semantic specification for modular addition because it has natural mathematical semantics, but bitvector arithmetic needs such specification.
>
> We agree we should elaborate on the semantics here. The semantics of “x & y” when interpreted mod $n$ is to first compute the bitwise operation assuming a base-2 representation, then interpret the resulting bitpattern as a number and reduce mod $n$. We will include this elaboration in future revisions.

---

### Official Review · Reviewer_c7mJ · 2022-10-25

**Confidence:** 5
**Correctness:** 4
**Technical Novelty And Significance:** 3
**Empirical Novelty And Significance:** 3
**Recommendation:** 5

**Clarity, Quality, Novelty And Reproducibility:**

The paper is clear and easy to follow. It is also novel and original in that the paper proposed a new theoretical notion to indicate that a programming language is embeddable.


**Strength And Weaknesses:**

Strengths:

+ The notion of equivalence-preserving program embedding is new, theoretically formulated, and proven to be satisfied by a simple arithmetic language and unsatisfied by a bit-vector language.

+ The usefulness of equivalence-preserving program embedding is experimentally demonstrated.

+ The documentation is well written and easy to follow.

Weaknesses:

- Little languages have equivalence-preserving program embedding functions. Theoretically, a tractably embeddable language (i.e., one with an equivalence-preserving embedding function) has a means to determine the equivalence of two programs. It means that no Turing-complete language is tractably embeddable. Furthermore, although recognizing the limitation of an approach is crucial in general, even the simple bit-vector language does not have such a function. Therefore, I am concerned about the practicality of the proposed approach in its current status. Considering the strictness of equivalence-preserving program embedding, I am not surprised that a machine learning model can learn an embedding for a tractably embeddable language even when the same model cannot learn an embedding for a language that is not tractably embeddable.

- As a minor point, the theory of the paper seems to assume the input space ($I$) is finite, but, in general, one can consider programs with an infinite input space (like list-processing programs).


**Summary Of The Paper:**

The paper proposed a new notion of equivalence-preserving program embedding, proved a simple arithmetic language has an equivalence-preserving embedding, and experimentally demonstrated that having the embedding is crucial to learn an embedding of programs.


**Summary Of The Review:**

The paper addresses a critical aim to apply machine learning techniques to programming language tasks. The proposed notion of equivalence-preserving program embedding is novel. However, as described above as a weakness, its practicality is a big concern because of its theoretical strictness. Research given in Section 5.1 would enhance the work much more.

---

> ### Author Response · Authors · 2022-11-18
> **Reviewer c7mJ Response**
>
> ## Reviewer c7mJ
>
> > Little languages have equivalence-preserving program embedding functions. Theoretically, a tractably embeddable language (i.e., one with an equivalence-preserving embedding function) has a means to determine the equivalence of two programs. It meNans that no Turing-complete language is tractably embeddable. Furthermore, although recognizing the limitation of an approach is crucial in general, even the simple bit-vector language does not have such a function. Therefore, I am concerned about the practicality of the proposed approach in its current status.
> Considering the strictness of equivalence-preserving program embedding, I am not surprised that a machine learning model can learn an embedding for a tractably embeddable language even when the same model cannot learn an embedding for a language that is not tractably embeddable.
>
> Formalizing tractability is an important contribution of our work. The predictions of the theory, the empirical results, and your expectation of the results all align. We believe this is evidence that our definitions accurately model the problem.

---

### Official Review · Reviewer_3JjL · 2022-10-26

**Confidence:** 3
**Correctness:** 3
**Technical Novelty And Significance:** 2
**Empirical Novelty And Significance:** 2
**Recommendation:** 3

**Clarity, Quality, Novelty And Reproducibility:**

The theoretical foundation presentation is clear. However, the potential of the idea is not thoroughly examined. The results should be reproducible.

**Strength And Weaknesses:**

(+) Code semantics is an important program property, and proposing a theoretical framework to incorporate semantics to embedding is important.

(+) The theoretical foundation is good.

(-) The demonstration is shown on small tiny problems.

(-) I am not sure any general-purpose language can be tractably embedded. It is not clear how such a definition can be extended to such languages.

(-) If the intention is to use tractable embedding to domain-specific language, I would encourage the authors to show some convincing scenarios.

**Summary Of The Paper:**

The paper proposes a theoretical background on how to incorporate code semantics into code embedding.  They argue that when two programs behave identically on a given input, they are semantically equivalent, and their representation in the embedded space should capture such properties. They demonstrate their theoretical foundation across two tasks.

**Summary Of The Review:**

The paper presents some interesting and important ideas. However, there is a large gap between the idea and practicality and generalizability.  It is not clear to me how the proposed idea of tractable embedding can be applied to real-world problems.

---

### Official Review · Reviewer_M5Qy · 2022-11-02

**Confidence:** 4
**Correctness:** 4
**Technical Novelty And Significance:** 3
**Empirical Novelty And Significance:** 3
**Recommendation:** 5

**Clarity, Quality, Novelty And Reproducibility:**

Clarity: the paper describes the core idea clearly.

Quality: the quality of the paper is good.

Novelty: the theory of equivalence-preserving embedding is novel.

Reproducibility: the evaluation is easy to reproduce.


**Strength And Weaknesses:**

Strength:
+ Describes an important property of program embeddings: they should remain invariant to semantic-preserving transformations.
+ First theoretical definition of equivalence-preserving program embedding problem. The paper provides formal conditions under which the programming languages can be tractably embedded.

Weaknesses:

- The empirical study is not convincing by only evaluating BERT-Tiny. Numerous neural architectures have been used to model programs, e.g., large language models, graph neural networks, etc. Can the proposed theory help explain some of the successes of one architecture over others? Can the theory guide how to develop new models to learn program representations?

- The practical implication of this paper is unclear. While the authors describe two applications (Section 2), these applications often deal with common programming languages that are intractable, e.g., code clones across binary code for vulnerability detection. Can the proposed theory help explain if the same code modeling task for some languages is strictly easier than the others? The theory can be more practically useful if it can be extended to quantify the intractability level so the resulting embeddings' error can be bounded or compared.

- Figure 1: unclear why certain input sizes have their accuracy going down, even though they have reached 100% in the earlier epochs.

- There is no discussion on why larger input space sizes need a smaller number of epochs to converge on the tractable language, which contradicts the observation that losses increase monotonically with input space sizes on the intractable language.
- Extensive results and discussions are put in the Appendix, costing great effort in going back and forth.

- The paper uses code clone detection and semantic labeling to motivate their theory, but the theory focuses on characterizing language tractability. Can the theory extend to cross-language clone detection, e.g., one language is tractable, but the other language is not?

**Summary Of The Paper:**

This paper defines equivalence-preserving program embedding, a constraint that requires the learned program embeddings to be equivalent if the corresponding programs have the same input-output behavior. It provides the conditions specifying when a programming language can be represented into embeddings that preserve the equivalence. It then provides empirical evidence showing that programming languages that satisfy the conditions can be more accurately embedded.

**Summary Of The Review:**

My main concern is the practical implication of the proposed theory. Ideally, equivalence-preserving program embedding makes sense if it remains invariant against program transformations. For example, code clone detection requires the model to understand that various classes of transformations on program syntax do not change their input-output behavior. Therefore, it makes more sense to characterize what transformations can be tractably embedded. However, the authors characterize it from the language perspective. Most of the programming languages that existing applications consider are intractable, i.e., assembly code or even undecidable, as pointed out by the authors.

Therefore, it is not unclear how the theory of equivalence-preserving embeddings could help solve existing semantic code modeling tasks.

---

> ### Author Response · Authors · 2022-11-18
> **Reviewer M5Qy Response**
>
> > The empirical study is not convincing by only evaluating BERT-Tiny. Numerous neural architectures have been used to model programs, e.g., large language models, graph neural networks, etc.
>
> Changing architectures or increasing model size would likely change the model performance. However, the asymptotic trends from our tractability analysis dictate the necessary model capacity as problem size scales. As a result, even for different architectures and larger models, we would expect poor performance on intractable languages.
>
> > Can the proposed theory help explain some of the successes of one architecture over others? Can the theory guide how to develop new models to learn program representations?
>
> While this is an interesting question, comparing architectures is outside of the scope of our paper.
>
> > Figure 1: unclear why certain input sizes have their accuracy going down, even though they have reached 100% in the earlier epochs.
>
> We investigated this behavior on 16 distinct training seeds (including the ones in our submission), and we found that only 2 of them demonstrate this sharp dropoff after reaching 100% accuracy. We believe this behavior is worthy of further investigation but out of scope for this paper.
>
> > There is no discussion on why larger input space sizes need a smaller number of epochs to converge on the tractable language, which contradicts the observation that losses increase monotonically with input space sizes on the intractable language.
>
> This does not contradict the observations in our paper. Our observations refer to the accuracy achieved after training is finished, as stated in the caption of Figure 3. Analyzing the complicated and poorly understood training dynamics of NNs is out of scope for this paper.
>
> > Extensive results and discussions are put in the Appendix, costing great effort in going back and forth.
>
> Thank you. We will work to improve the presentation of our empirical results.
>
> > The paper uses code clone detection and semantic labeling to motivate their theory, but the theory focuses on characterizing language tractability. Can the theory extend to cross-language clone detection, e.g., one language is tractable, but the other language is not?
>
> If the intermediate language used in [Pewny et al. 2015](https://ieeexplore.ieee.org/document/7163056) is tractable and the mapping from the languages to the intermediate language is sufficiently fast (polynomial time in program size and polylogarithmic time in input space size), then the languages are tractable. The existence of such a mapping to the intermediate language for a tractable and intractable language is outside of the scope of this paper.
>
> > Ideally, equivalence-preserving program embedding makes sense if it remains invariant against program transformations. For example, code clone detection requires the model to understand that various classes of transformations on program syntax do not change their input-output behavior. Therefore, it makes more sense to characterize what transformations can be tractably embedded. However, the authors characterize it from the language perspective.
>
> This is an interesting idea, but it is a separate research direction outside the scope of this paper.

---

### Author Response · Authors · 2022-11-18
**General Response**

We thank all reviewers for their feedback. We address concerns common to multiple reviewers in this comment and respond to individual reviewers separately.

## Technique vs. Theoretical Characterization (Reviewers 3JjL, c7mJ)

We first address what we believe is a misunderstanding of the contribution of the paper.

> [Reviewer 3JjL]: It is not clear to me how the proposed idea of tractable embedding can be applied to real-world problems.

> [Reviewer c7mJ]: The paper proposes a theoretical background on how to incorporate code semantics into code embedding. They argue that when two programs behave identically on a given input, they are semantically equivalent, and their representation in the embedded space should capture such properties.

> [Reviewer c7mJ]: although recognizing the limitation of an approach is crucial in general, even the simple bit-vector language does not have such a function. Therefore, I am concerned about the practicality of the proposed approach in its current status.

We emphasize that we do not present a new embedding technique. Our paper is about theoretically characterizing and empirically validating the difficulty of the equivalence-preserving embedding problem. To that end, we provide hardness results applicable to existing embedding techniques in the literature. We advocate for the view that learning semantics of programming languages is challenging and that researchers should think carefully about the semantic properties their models predict.

## Applicability (Reviewers M5Qy, c7mJ)

We address the relevance of our theory to programming languages used in practice.

> [Reviewer M5Qy]: My main concern is the practical implication of the proposed theory ... Most of the programming languages that existing applications consider are intractable, i.e., assembly code or even undecidable, as pointed out by the authors.

> [Reviewer c7mJ]: The proposed notion of equivalence-preserving program embedding is novel. However, as described above as a weakness, its practicality is a big concern because of its theoretical strictness.

General-purpose languages such as Python and C++ are intractable. This holds no bearing on the quality of our tractability analysis. It is a reflection of the difficulty of the equivalence-preserving embedding problem, which many program embedding techniques attempt to solve (Pewny et al., 2015; Hu et al., 2017; Yu et al., 2019; Mou et al., 2016; Ben-Nun et al., 2018; Puri et al., 2021; Wang et al., 2018).

Our result suggests that for a neural network to perfectly solve large-scale semantic reasoning tasks, the target language must be tractable. For example, producing equivalence-preserving embeddings for even simple languages, such as bitvector arithmetic, can be intractable.

## Gradations (Reviewers M5Qy, Pnio)

A common concern was that our theory does not provide enough gradations to characterize the difficulty level of tasks beyond tractable or intractable.

> [Reviewer M5Qy]: Can the proposed theory help explain if the same code modeling task for some languages is strictly easier than the others? The theory can be more practically useful if it can be extended to quantify the intractability level so the resulting embeddings' error can be bounded or compared.

> [Reviewer Pnio]: For example, for existing programming semantic tasks, can the authors classify their hardness using the theory developed in this paper?

In our paper, tractability is phrased in terms of the asymptotic behavior of embedding time and embedding size. Rather than classifying a language as tractable or intractable, we can compare the asymptotic growth of both embedding time and embedding size, providing gradations of tractability.

---

> ### Author Response · Authors · 2022-11-18
> **General Response (Cont.)**
>
> ## Empirical Study Scale (Reviewers 3JjL, c7mJ, Pnio)
>
> A common concern was that the scale of our empirical study was not large enough.
>
> > [Reviewer 3JjL]: The demonstration is shown on small tiny problems.
>
> > [Reviewer c7mJ]: Considering the strictness of equivalence-preserving program embedding, I am not surprised that a machine learning model can learn an embedding for a tractably embeddable language even when the same model cannot learn an embedding for a language that is not tractably embeddable.
>
> > [Reviewer Pnio]: The evaluation is rough: The evaluations are on two toy languages defined in this paper, and the input space size is at most 11. It is conceivable that modular addition is easier than bitvector arithmetic to learn because the latter is a direct extension of the former. It is inconclusive to understand model's capacity to learn the semantics just based on this toy evaluation.
>
> The goal of our empirical study is to show our theory captures the difficulty of the problem by showing it is predictive of real phenomena. In particular, we consider a tractable language and extend it to include bitwise logical operators. Our theory predicts that even this small extension produces a language that is not tractably embeddable, which we confirm empirically.
>
> We argue that considering a larger, general-purpose language would not lead to additional insight into the predictive power of our theory. Both languages we consider can be embedded as subsets of Python, for example. Thus, supersets of the intractable language (e.g., Python) are provably intractable, and we expect to see a similar trend in the asymptotic behavior.
>
> ## Finite Input Space (Reviewers c7mJ, Pnio)
>
> Finally, we address the concern that we only consider finite input spaces.
>
> > [Reviewer c7mJ]: As a minor point, the theory of the paper seems to assume the input space (I) is finite, but, in general, one can consider programs with an infinite input space (like list-processing programs).
>
> > [Reviewer Pnio]: These are naturally encountered formal systems. In this paper, the authors impose assumptions like the (denotational) semantics is from a finite set, the input space is finite. Those assumptions in general do not hold.
>
> Considering infinite input space sizes complicates our theoretical development with marginal benefit. Our theory shows an equivalence-preserving embedding function exists for a language exactly when there are finitely many semantic equivalence classes. We are not aware of any non-contrived languages with infinite input spaces that have finitely many equivalence classes [1]. Thus, to our knowledge, equivalence-preserving embedding functions do not exist for most practical languages with infinite input spaces.
>
> With that being said, we could extend our theory to detect when a language with an infinite input space has finitely vs. infinitely many equivalence classes as follows. Parameterize the semantics by the input space size (e.g., the number of unique lists that list processing programs are subject to). If for every input space size $n$, there are $f(n)$ semantic equivalence classes, then an equivalence-preserving embedding function exists exactly when $f(n)$ is upper bounded by a constant.
>
> We prefer to omit these technical extensions from the paper, as we do not believe they significantly extend the applicability of the theory.
>
> [1] A contrived example of a language with an infinite input space but finitely many semantic equivalence classes is a language with a single expression $x$ with an input space over the integers. This language has a single semantic equivalence class, no matter the size of the input space.

---

### Decision · Program_Chairs · 2023-01-20

**Decision:**

Reject

**Justification For Why Not Higher Score:**

Theory makes assumptions that significantly reduce its applicability to more general purpose programming languages.

Empirical study shows results on toy programming languages and small representation learning models.

Limited insights for improving the current program embedding pipelines.

**Justification For Why Not Lower Score:**

Can't be lower.

**Metareview: Summary, Strengths And Weaknesses:**

This paper presents an interesting characterization of equivalence-preserving program embeddings and studied under what conditions tractable embeddings exist for a programming language.  While the conceptual framework seems interesting, the assumptions required by the framework and the empirical studies show quite limited applicability to more general-purpose widely used programming languages, which all reviewers are concerned about.  As a theory paper this work also doesn’t discuss any insights for, e.g. learning program embeddings better.  I still think the problem of equivalence-preserving program embeddings is a good one to study, but the authors can try to think more along the above directions and improve the paper further for the machine learning community.